# Polymorphic Malignant Melanoma (PMM) of the Left Helix: Case Report with Clinical-Pathological Correlations

**DOI:** 10.3390/diagnostics12112713

**Published:** 2022-11-06

**Authors:** Gerardo Cazzato, Anna Colagrande, Michele Maruccia, Eleonora Nacchiero, Carmelo Lupo, Nadia Casatta, Giuseppe Ingravallo, Eugenio Maiorano, Andrea Marzullo, Giuseppe Giudice, Leonardo Resta

**Affiliations:** 1Section of Molecular Pathology, Department of Emergency and Organ Transplantation (DETO), University of Bari “Aldo Moro”, 70124 Bari, Italy; 2Section of Plastic Surgery, Department of Emergency and Organ Transplantation (DETO), University of Bari “Aldo Moro”, 70124 Bari, Italy; 3Innovation Department, Diapath S.p.A., Via Savoldini n. 71, 24057 Martinengo, Italy

**Keywords:** polymorphic, malignant melanoma, neoplasm, melanocytic, skin, immunohistochemistry, differential diagnosis

## Abstract

Malignant melanoma (MM) is known to be the great mimic in dermatopathology. Over time, several variants have been described, not all of which have repercussions on the clinical/oncological management of the affected patient. The existence, however, of these alternative forms of MM is of great interest to the pathologist, as they are potentially capable of inducing diagnostic errors affecting the diagnostic-therapeutic care pathway (PDTC). In this paper, we present a very rare case of polymorphic MM, in which five different morphological aspects coexisted in the same lesion, confirmed by immunohistochemical investigation and by RT-PCR for mutation of the BRAF gene and discuss the importance of correct recognition of these different morphological features to avoid misdiagnosis.

**Figure 1 diagnostics-12-02713-f001:**
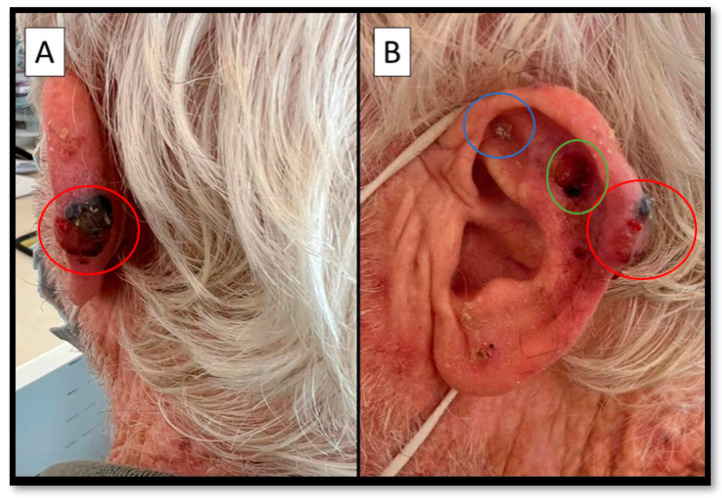
(**A**,**B**) Posterior and lateral view of an infiltrated nodule, hard on palpation, black and surmounted by a large ulcerative-necrotic area (red circle). In (**B**) two other, different, lesions are apparent (blue and green circles). A 73-year-old man presented at the Plastic Surgery Complex Operative Unit because of the appearance of a lesion that was present from more than two years before, at the level of the left helix, measuring about 18 × 12 mm (red circle, Figure 1), with an ulcerated area on the surface. The medical history was negative for other pathologies and/or neoplasms. After a plastic surgery specialist consultation, the decision was made to remove the lesion together with two other neighboring lesions (blue and green circles, Figure 1B). The samples were histopathologically analyzed.

**Figure 2 diagnostics-12-02713-f002:**
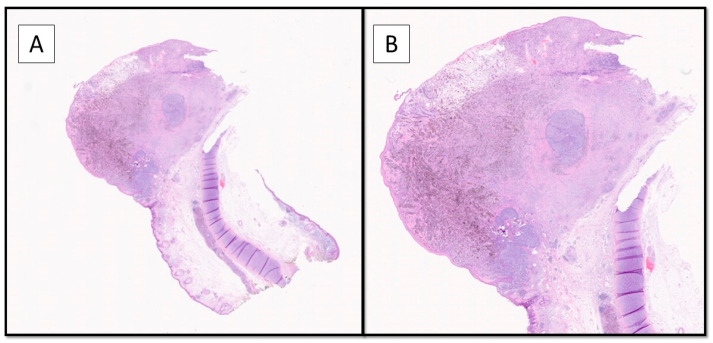
The histopathological evaluation showed a nodular lesion overlaying the auricular cartilage and extending to ulcerate the overlying epidermis (Hematoxylin-Eosin, Original Magnification 2× (**A**) and 4× (**B**)). A diagnosis of polymorphic malignant melanoma (PMM) was made, with a vertical growth phase of tumorigenic type, up to 4.5 mm thick according to Breslow, with a number of mitoses/mmq> 1, as well as lymphovascular invasion, neurotropism, and microsatellitosis, staged according to the AJCC 8′Edition as pT4b N1c. The sample was then subjected to molecular biology tests to search for mutations in exons 11 and 15 of the BRAF gene and in exons 2, 3, and 4 of the NRAS gene by RealTime PCR. This revealed a mutation in codon V600 of exon 15 of the BRAF gene (V600E/E2/D). No mutations were detected in exons 2, 3, and 4 of the NRAS gene.

**Figure 3 diagnostics-12-02713-f003:**
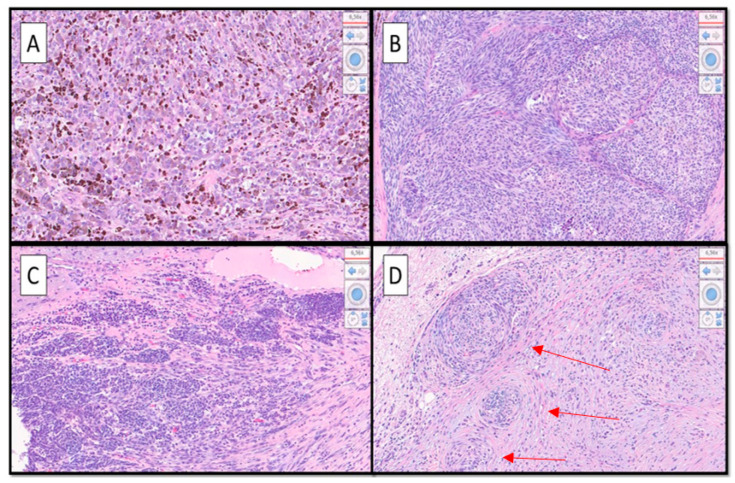
(**A**–**D**). At higher magnification, a heterogeneous morphological pattern was evident, consisting of different aspects; alongside the superficial spreading type (SSM) melanoma component (not shown) there were hyperpigmented epithelioid cell aspects (**A**), hypopigmented spitzoid features (**B**), nevoid aspects that tended to lose HMB-45 immuno-expression (**C**), and aspects of desmoplastic/neurotropic melanoma (**D**). Furthermore, a rhabdoid appearance could be appreciated (not shown). Note in (**D**) neurotropism of malignant melanoma cells (red arrows). (Hematoxylin-Eosin, Original Magnification 20×). Indeed, it has been decided to have the patient undergo sentinel lymph node surgery (BLS) and then, based on the result, a decision will be made for possible medical therapy. It is very important to recognize the different morphological forms of MM, as it is recognized in the literature to what varying extents this neoplasm is able to be “plastic”. For example, in a recent paper, Massi D. et al. dealt with the discourse relative to the morphophenotypic and genetic plasticity of MM, analyzing the very rare case of dedifferentiated MM [1].

**Figure 4 diagnostics-12-02713-f004:**
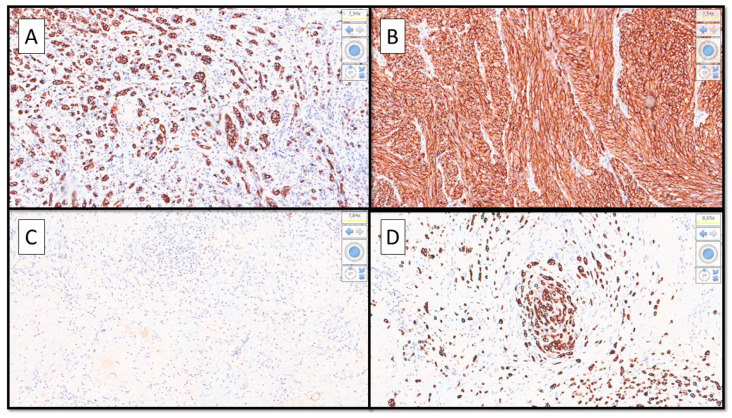
(**A**–**D**). (**A**) Immunohistochemical preparation with anti-Human Melanoma Black Antigen-45 (HMB-45) antibody showing strongly positive epithelioid cell features. (**B**) Immunopositivity for HMB-45 in the spitzoid part of the lesion. (**C**) Almost entirely negative immunoreaction for HMB-45 in the nevoid part of the polymorphic malignant melanoma. (**D**) Detail of the desmoplastic/neurotropic part of the MM infiltrating a small nerve. (Immunostaining for HMB-45, Original Magnification 10× (**A**–**C**) and 20× (**D**). The case presented is of particular interest for three reasons: (1) Malignant melanoma is plastic, both from a morphological/histological point of view and from a molecular point of view [2,3]. (2) The knowledge of different morphological variants of MM is of fundamental importance to be able to correctly diagnose this entity [4]. (3) It is essential to implement basic and clinical research in an attempt to better understand the molecular events underlying the histological heterogeneity and the potential impact on patient prognosis [5,6].

## Data Availability

Not applicable.

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
