# Peer review of "Polymorphic Malignant Melanoma (PMM) of the Left Helix: Case Report with Clinical-Pathological Correlations"

_diagnostics, 2022, doi:10.3390/diagnostics12112713_

Round 1

Reviewer 1 Report

1. Please invite a Pathologist who is a primary English-language speaker as a co-author to rewrite the manuscript in plain and standard medical English. In it's present form the grammar is so circuitous that it is almost unreadable. 

2. In the last paragraph you state that 'there are 3 reasons", but you only list 2! 

Author Response

Dear Reviewer n'1,

first of all, thank you very much for your suggestions. We have corrected the entire manuscript by a native English-language and we have added the third reason.

A warm greeting

Reviewer 2 Report

The "Interesting image" by Cazzato and colleagues presents a nice clinical case of Polymorphic Malignant Melanoma (PMM) affecting a middle-age man on his left helix. The case is well documented regarding the clinical and histologic images, but I suggest to add the demoscopic features (maybe they were consistent with melanoma, although I believe not sufficient to suspect the polymorphic entities, of course).  I believe dermoscopy is a need for dermatology readers. 

I am convinced that the presentation should be re-ordered, because the paper is too short, with the Figure legends very long (they seems to a be a part of the manuscript and not only Fig legends). The case is quite rare, suggestive and nice, but should be kept more appealing!!!

The case misses the "History conclusion", that means which treatment was decided after surgery (and was the SNB performed???), maybe target therapy was started, but it should be stated. What about the history of patient, he is still alive or not? Since there are not restrictions on the length of the paper (see Author guidelines), this should be reported.

Please also add a short discussion on other previously reported similar cases (also see the Refs).

Lastly, English should be checked by a native English speaker.

Author Response

Reviewer n’2: The "Interesting image" by Cazzato and colleagues presents a nice clinical case of Polymorphic Malignant Melanoma (PMM) affecting a middle-age man on his left helix. The case is well documented regarding the clinical and histologic images, but I suggest to add the demoscopic features (maybe they were consistent with melanoma, although I believe not sufficient to suspect the polymorphic entities, of course).  I believe dermoscopy is a need for dermatology readers.

Answer n’1: Dear Reviewer n’2, first of all thank you very much for these beautiful comments. We haven’t dermoscopic images because the dermatologist that have studied, firstly, the lesion was external. We hope that it will be not a great problem. Thanks so much.

Reviewer n’2: I am convinced that the presentation should be re-ordered, because the paper is too short, with the Figure legends very long (they seems to a be a part of the manuscript and not only Fig legends). The case is quite rare, suggestive and nice, but should be kept more appealing!!!

Lastly, English should be checked by a native English speaker.

Answer n’2: Dear Reviewer n'2, thank you very much. We subjected the manuscript to an extensive check by a native English speaker and, on this occasion, we tried to modify some aspects of the paper. However, we have not been able to upset the order as the rules of Diagnostics such as "Interesting Image" are just this type. A hug

Reviewer n’2: The case misses the "History conclusion", that means which treatment was decided after surgery (and was the SNB performed???), maybe target therapy was started, but it should be stated. What about the history of patient, he is still alive or not? Since there are not restrictions on the length of the paper (see Author guidelines), this should be reported.

Answer n’3: Dear Reviewer n’2, thank you. We added these important informations.

Reviewer n’2: Please also add a short discussion on other previously reported similar cases (also see the Refs).

Answer n’4: Dear Reviewer n’2, done. Thank you.

Round 2

Reviewer 1 Report

(none)

Reviewer 2 Report

The Authors have addressed the required issues.